## Research Article

schizophrenia; relapse prevention; clinician perspectives; barriers to care; treatment - non-adherence

**Corresponding author:**
Anna Margaretha Smit;
Email: amsmit@sun.ac.za

# Beyond treatment non-adherence: A qualitative study of clinicians' perspectives on structural and social determinants of schizophrenia relapse in South Africa

Anna Margaretha Smit[1] , Sanja Kilian[2], Hilmar Klaus Luckhoff[2], Lebogang Simon Phahladira[2], Robin Emsley[1] and Laila Asmal[2]

[1]Psychiatry, Stellenbosch University Faculty of Medicine and Health Sciences, South Africa and [2]Stellenbosch University Faculty of Medicine and Health Sciences, South Africa

## Abstract

The objective of this study is to explore public sector clinicians' perspectives on factors associated with relapse in schizophrenia within a South African context, focusing on structural, social and environmental contributors beyond treatment non-adherence. Three focus groups were conducted with 14 public-sector clinicians (psychiatrists, medical officers, psychiatric registrars and psychiatric nurses) with ≥5 years' experience in schizophrenia care. Data were analysed using reflexive thematic analysis to identify themes relating to relapse risk. Clinicians consistently described medication non-adherence, often the immediate trigger for relapse, as emerging from interrelated health system and socio-structural constraints, including poverty, unemployment, unsafe communities, fragmented services and stigma. Limited access to newer-generation antipsychotics, medication stockouts, early discharges due to bed shortages and scarce post-discharge rehabilitation compounded relapse risk. Family support was frequently undermined by financial strain and competing demands, while crime and gang violence discouraged clinic attendance. Stigma within both communities and healthcare settings reduced trust and engagement. In this lower-middle-income country context, relapse prevention depends on integrated strategies that combine clinical management with interventions addressing structural and social determinants. Policy priorities include strengthening primary-level mental healthcare, ensuring medication supply continuity, expanding supervised care and vocational programmes, implementing stigma-reduction initiatives and fostering intersectoral collaboration to address safety and spatial inequities in service provision.

## Impact statement

Relapse in schizophrenia remains a major challenge in lower-middle-income countries (LMICs), where mental health services are often overstretched and under-resourced. While relapse is frequently explained in terms of individual behaviour, particularly treatment non-adherence, this perspective overlooks the wider social and system-level barriers that shape people's ability to stay well. This study helps close that gap by foregrounding the voices of frontline clinicians who work daily with individuals living with schizophrenia, yet whose insights are underrepresented in research or policy discussions. Our findings show that clinicians view relapse not as a single-cause event, but as the outcome of many interacting factors. They describe how clinical vulnerabilities intersect with everyday social pressures, economic hardship, stigma and health-system constraints. Their accounts highlight how the environment in which care is delivered, such as the availability of staff, continuity of services, community follow-up and support structures, plays a major role in shaping relapse trajectories. These insights have important implications for practice and policy. Understanding relapse through a broader contextual lens may help inform more responsive service planning, strengthen primary care involvement and support the development of relapse-prevention strategies that reflect real-world challenges. This knowledge is valuable for clinicians, managers, policymakers, community mental health workers and families to better support individuals living with schizophrenia. Although grounded in a South African setting, the findings speak to wider issues facing mental health systems across many LMICs. By highlighting clinicians' accounts of how social and structural constraints shape clinical care, this study contributes descriptive, context-specific insights that may inform ongoing discussions about the development of more equitable and contextually appropriate mental health services.

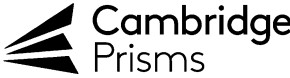

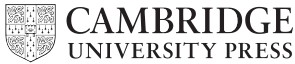

## Introduction

### Background

Schizophrenia is a chronic, debilitating disorder, with approximately one-third of individuals relapsing within the first year after a first episode (Adebiyi et al., 2018) and many experiencing recurrent episodes across the illness course (Robinson et al., 1999). Relapse carries significant psychosocial consequences (Kane, 2007) and biological sequelae, including progressively reduced treatment responsiveness (Emsley et al., 2013). Quantitative research consistently identifies treatment non-adherence as a central predictor of relapse (Robinson et al., 1999). Additional correlates include trauma histories, poor premorbid adjustment and reduced quality of life (Turkington et al., 2009; Alvarez-Jimenez et al., 2012; Boyer et al., 2013; Petros et al., 2021; Tosato et al., 2021). However, while these studies identify risk factors, they provide a limited understanding of how relapse develops in routine care, particularly in lower-middle-income countries (LMICs) marked by resource constraints and social adversity. A recent systematic review reported that nearly half of individuals with schizophrenia in sub-Saharan Africa were not adherent to treatment, primarily due to substance misuse, socio-cultural barriers and antipsychotic side effects (Addisu et al., 2025). Delays in dispensing Clozapine for the estimated 30% with treatment-resistant schizophrenia (Iqbal et al., 2021) have also been identified as critical service failures (Whiskey, 2021).

Qualitative studies examining how these risk factors operate in practice remain limited. Clinical perspectives are particularly valuable for understanding the interaction between relapse, health systems and care delivery (Ahmad et al., 2014). Research from Ethiopia (Teferra et al., 2013) and Thailand (Intharit et al., 2021) highlights biopsychosocial contributors, including non-adherence, limited social support, substance misuse, stigma, unrealistic expectations of cure, medication side effects, perceived inefficacy, restricted access to first-generation antipsychotics, stressful life events, comorbidities, pregnancy-related changes and strained service user–provider relationships. In South Africa, psychiatric nurses in KwaZulu-Natal identified cultural beliefs, poor family support and substance misuse as barriers to adherence (Nxasana and Thupayagale-Tshweneagae, 2014), while occupational therapists in Johannesburg emphasised limited social participation and poor community integration as drivers of relapse and readmission (Smith et al., 2014).

Clinicians possess critical insight into systemic gaps, although hierarchical structures and fear of repercussions may inhibit open reflection (Kish-Gephart et al., 2009; Okuyama et al., 2014). In South Africa, ~80% of the population depends on the public healthcare system, which allocates about 5% of its budget to mental health (Docrat et al., 2019). Under the Mental Health Care Act (No. 17 of 2002), individuals with acute psychosis are admitted to district hospitals for 72-h observation and stabilisation before referral to specialist services when indicated. Post-discharge follow-up occurs in primary healthcare clinics, where services are poorly integrated and largely delivered by generalist nurses with limited psychiatric training (Sorsdahl et al., 2023). This tiered structure contributes to discontinuities, particularly during hospital-to-community transitions, where follow-up and psychosocial support are minimal. An international consensus review found that relapse was defined solely by clinical judgement in 85% of studies, without standardised criteria (Howes et al., 2025), emphasising the subjective nature of relapse determination and the importance of qualitative inquiry in resource-constrained contexts.

This study aimed to move beyond the dominant focus on non-adherence by examining broader contributing factors within resource-constrained settings. As the first phase of a multi-stakeholder investigation, it establishes a foundation for identifying system-level barriers and exploring how these intersect with lived experience.

## Methodology

### Participant recruitment and study design

First author RS and co-author HL conducted three focus groups with purposively selected clinicians with at least 5 years' experience in schizophrenia care. The first group comprised five psychiatrists from Stikland Psychiatric Hospital; the second included one psychiatrist and two psychiatric nurses from community clinics; and the third consisted of six medical officers from Lentegeur Psychiatric Hospital in Cape Town's northern suburbs. Department heads facilitated recruitment. Groups were limited to six participants to promote active engagement, as larger groups may hinder discussion (Hennink et al., 2019). Sessions were scheduled according to availability.

### Procedure and data collection

A semi-structured interview guide, informed by social constructivist theory and empirical literature, facilitated dynamic discussion (McWilliam et al., 2009). Broad questions allowed participants to prioritise salient issues, with probes exploring individual, service-level and socio-structural factors. The semi-structured interview guide is attached as Supplementary Table S2. This approach enabled participants to draw on prior knowledge and construct meaning, generating deeper insight (Hutchinson and Huberman, 1994) to inform education and policy (Thomas et al., 2011; Andrews, 2012). Focus groups were conducted in English, audio-recorded, supported by field notes, lasted 40–50 min and each group met once.

### Research positioning

My interest in clinicians' perspectives on relapse stems partly from over 20 years' experience as a psychiatric nurse and researcher at an academic tertiary institution in South Africa, focusing on schizophrenia's clinical features and pharmacological treatment. Conducting multiple clinical studies within a well-resourced environment shaped a controlled, individual-level understanding of relapse centred on adherence, treatment response and symptom management.

In contrast, public-sector clinicians often work in resource-constrained settings with low provider-to-client ratios and limited infrastructure. In my research role, I had autonomy to schedule visits, access to various consultation rooms, provide travel reimbursement and refreshments, ensure medication supply and provide adherence monitoring and psychoeducation, conditions that controlled extraneous variables.

During data collection, clinicians described working within significant structural constraints. While non-adherence and substance use were noted, poverty, stigma, crime and systemic under-resourcing were emphasised as key disruptions to continuity and client agency. These discussions highlighted how well-resourced research settings may underestimate social determinants' influence on outcomes. Although reflecting the first author's perspective, the analysis was strengthened by co-authors SK, LP and LA, whose public-sector expertise helped interrogate assumptions linked to a research-oriented lens.

### Data analysis

Data were analysed using reflexive thematic analysis following Braun and Clarke's (2019) six-phase approach. Audio recordings were transcribed verbatim in Microsoft Word, with repeated listening to ensure accuracy. RS and WM independently familiarised themselves with transcripts through repeated reading and analytic notes, maintaining reflexive journals to document interpretations, assumptions and emotional responses, which were discussed in analytic meetings to enhance reflexivity and transparency.

Initial coding involved systematically generating concise, meaningful labels relevant to the research question. RS and WM coded independently using structured Excel spreadsheets to organise participant identifiers, extracts, codes and notes. Coding was iterative, with codes refined, merged or discarded as patterns emerged. Reflexive discussions explored interpretive differences and informed analytic decisions. Codes were then reviewed for conceptual similarity and organised into candidate sub-themes and themes, collapsing overlaps and excluding irrelevant extracts. RS and WM conducted initial coding; SK and LA joined from the thematic review stage. The thematic framework was reviewed collaboratively, ensuring internal coherence, clear distinctions and alignment with study objectives.

In the final stage, RS, SK and LA undertook deeper interpretive analysis to clarify theme boundaries and meanings. Illustrative extracts were selected collaboratively, and theme names were refined for conceptual clarity. The final thematic structure reflected interrelationships between individual, structural and social contributors to relapse.

### Results

#### Demographic data of participants

The sample included 14 healthcare professionals (3 males and 11 females): 2 psychiatric nurses, 4 psychiatrists, 2 psychiatric registrars and 6 medical officers, with a mean age of 40 years. Eleven were based at psychiatric hospitals, including both inpatient and outpatient facilities, and three in community mental health clinics. No gender-based differences in clinicians' perspectives were identified, though this should be interpreted cautiously given the small, predominantly female sample. Differences did emerge by setting. Community clinic-based clinicians reported more frequent medication stockouts, transport barriers, clinic avoidance due to crime, clinic-related stigma and limited access to psychoeducation, vocational services and trained mental health staff. Hospital-based clinicians highlighted pressure for premature discharge due to bed shortages and the lack of safe or supported discharge options. Structural constraints such as staffing shortages and limited access to second-generation antipsychotics were noted across both settings.

Field notes indicated that hospital-based clinicians managed caseloads of 30–35 mental health clients at a given time, while clinicians based at the community clinics consulted ~147 individuals a day, with significant staff shortages contributing to high cognitive demands and frequent interruptions. Full participant details are provided in Supplementary Table S1.

### Main and subthemes identified

Participants acknowledged medication non-adherence as the leading cause of relapse, framing it within intersecting challenges rather than individual failings. (Figure 1). Three overarching themes emerged: (i) individual and treatment-related contributors; (ii) structural- and environmental-level contributors and (iii) social- and community-level contributors (Table 1).

### Individual and treatment-related contributors to relapse

#### Medication-related challenges

While non-adherence was often the immediate trigger for relapse, participants described external factors shaping treatment adherence, including poor follow-up support, short consultation times, fragmented care and medication that is limited to: *"the oldest of the old that has the worst of the worst side effects."* (A#01)

Some clients stopped treatment when they felt better: *"They're not even long out of hospital... the minute they start feeling better, they feel... 'I don't need to take my medication', not knowing that it's*

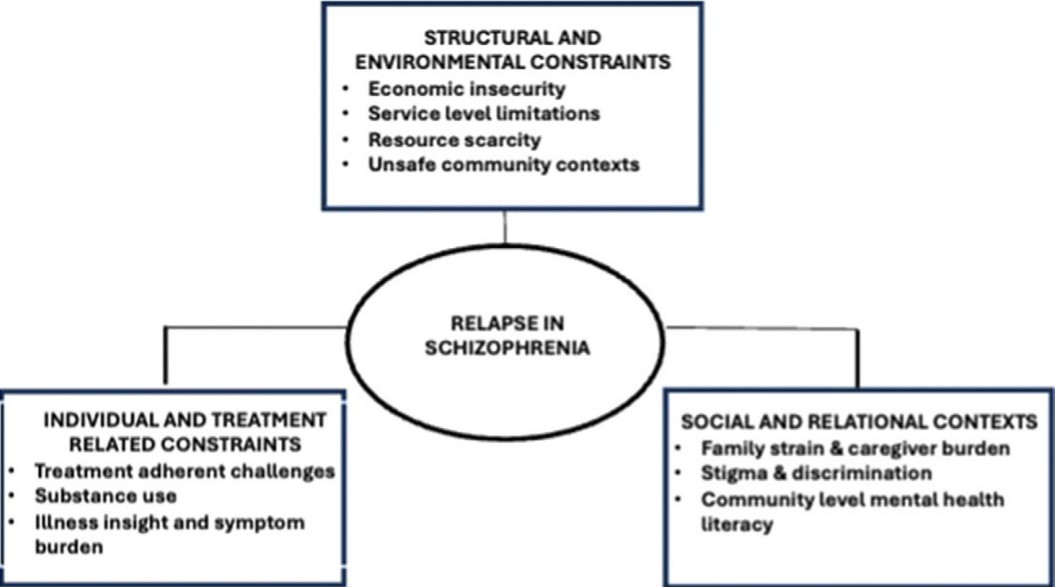

**Figure 1.** Conceptual framework of intersecting contributors to schizophrenia relapse.

**Table 1.** Themes and subthemes

| Main themes | Subthemes |
|---|---|
| Individual and treatment-related contributors to relapse | - Medication-related challenges<br>- Substance use and peer pressure |
| Structural and environmental contributors to relapse | - Poverty and economic insecurity<br>- Healthcare system shortfalls<br>- Crime and unsafe communities |
| Social and community-level contributors to relapse | - Family-related challenges<br>- Stigma, discrimination, and low mental health awareness |

*because of the medication that you are actually taking that you are feeling better.” (B#07)*

### Substance use and peer pressure

Substance use was described as both a coping mechanism and a barrier to treatment adherence, often reinforced by peer norms and rooted in unemployment, boredom and lack of recreational outlets:

*…especially with the younger crowd, also just having the wrong peer group…plays a big role concerning substance misuse. (C#12)*

But this is the home that they come from, and they're watching people who don't have a mental illness live a certain lifestyle. And then we're expecting them somehow to be different. (B#08)

It's very complex. Yes, we have to simplify by saying that it's substances, but…people live in poverty… people do not have any recreational anything to keep them busy people, do not have jobs. (B#08)

## Structural- and environmental-level contributors to relapse

### Poverty, economic insecurity and limited access to basic needs

Housing, food insecurity, a lack of supervision and transportation barriers were identified as factors that undermine continuity of care: *“Access to basic things that you need as a human being…access to basic housing…that's going to keep you from the streets…the lack of safety…people are not safe within their own homes… accessing help … depends on which area that you're in.” (B#08)*

Patients that have easy access to health facilities, they do better. (C#011)

While the South African government provides disability grants, bureaucratic challenges limit access to financial support:

Some of them don't have a disability grant…I think on the side of the government there isn't any help to get patients to the clinics. (C#010)

### Healthcare system shortfalls

Gaps in primary care integration, staff shortages, early discharges and limited access to rehabilitation services were mentioned as contributors to relapse:

*Mental health is not well integrated at primary care level. There is a kind of thought that a mental health client can only really be seen by a mental health nurse or by the registrar (resident) who reaches out once a week to the clinic. That of course is not true for any other patient who presents at the clinic. (A#01)*

We discharge earlier due to availability of beds… patients are not really ready to go. (C#10)

We have limited vocational type or psychosocial rehabilitation facilities. Drop-in centres or day programs….there are none in the community….They are really struggling to integrate again (A#03)

Medication shortages often occur, and service users may be turned away, asked to go to another facility or given substitutes with poor tolerability: *“they don't get the medication…they get something else. The patient maybe not happy about it or suffer side-effects…then they don't go.” (C#011).*

Discharge into environments saturated with stress and substance misuse contributes to relapse risk:

we often have to send patients back to settings where they are not supported, exposed to stresses and substances, sometimes even having to send patients to shelters because there's just limited support and limited options available… that also drives some of that therapeutic sense of futility and failure…. and it definitely drives the revolving door. (A#03)

A recent cybersecurity breach caused delays in laboratory results, which affected clozapine dispensing: *“if the pharmacy doesn't have the white cell counts…they can't dispense the medication.” (C#014)*

### Crime and unsafe communities

High crime rates, violence and gang activity in certain communities were noted as deterrents to clinic attendance: *“patients can't go to the clinic because they're scared…the clinic is in a dangerous area.” (C#09)*

*Not like a chronically dangerous area that the patient or the family are like wary of going to the clinic by foot, or currently, like a gang situation so, they can't go during that time. (C#14)*

Chronic exposure to crime, trauma and social instability creates ongoing stress that is not adequately addressed by traditional psychoeducation and stress reduction approaches aimed at preventing relapse: *“that's sometimes the most difficult thing to combat… we can't protect patients against those (life stressors) and our patients, given their circumstances, are invariably exposed to a lot of stress, trauma and crime.” (A#03)*

## Social- and community-level contributors to relapse

### Family-related challenges

Families were crucial to care but often overburdened by limited psychoeducation, financial strain and competing responsibilities: *“sometimes there isn't even a family member that they can go to… and not having that kind of support plays a big role.” (B#07)*

Many families face extreme financial hardship, which may push them towards relying on informal income sources to survive: *“the drug lord will give the people money to keep their stock at their house… if you don't have money and you can't buy food and the children are hungry, that's the easy way out to get money… it's a very complex thing.” (B#06)*

Families also contend with significant household stressors, such as job loss, illness and death in the family, which participants explained exacerbated instability in people with schizophrenia: *“stressors of family members passing away, losing jobs. I had a patient who lost 6 family members in a space of a year…a big stressor.” (C#013)*

### Stigma, discrimination and barriers to mental health access

Misconceptions about schizophrenia lead to exclusion, social ridicule and delayed treatment-seeking, contributing to a relapse of psychosis: *“The name calling in the community*: 'Have you taken your 'mal pille'?” [Afrikaans translated to 'crazy pills'] (B#06)

Stigma was also present within healthcare settings: *“the minute they find out you have a label of being a mental health patient, they overlook everything else”. (B#07)*

Mental health is often deprioritised in overburdened systems, contributing to poor health-seeking behaviour, the gateway for a relapse of psychosis: *"Mental health care is not everyone's priority… if whatever illness you have is placed at the back of the list by the government, by the people who manage healthcare services… there is a lot of limitations…specifically for mental health care users' with illnesses like chronic lifelong schizophrenia"*. (B#08)

## Discussion

This qualitative study explored clinicians' perspectives on relapse factors in schizophrenia to consider broader structural, social and environmental contexts. Three interlinked categories emerged: (i) individual- and treatment-related contributors; (ii) structural- and environmental-level contributors and (iii) social- and community-level contributors.

Although medication non-adherence was identified as the primary driver of relapse, clinicians situated it within structural constraints rather than individual failings. In LMICs, adherence is shaped by health system limitations and social determinants beyond personal choice (Jester et al., 2023). Participants described reliance on first-generation antipsychotics with significant side effects due to restricted access to second-generation antipsychotics, alongside limited capacity to monitor or manage side effects, factors known to undermine adherence (Boydell et al., 2003; Moritz et al., 2013; Gründer et al., 2016). Discontinuation when feeling better was understood not only as limited insight, but as attempts to reclaim normality and resist a medicalised identity (Strauss et al., 1985; Freudenreich et al., 2004), particularly in high-stigma, low mental health literacy settings.

Structural barriers, including poverty, unstable housing and transport difficulties, align with LMIC literature, identifying social adversity as a determinant of mental health outcomes (Lund et al., 2018; Patel et al., 2018). In South Africa, persistent spatial inequality and uneven resource distribution intensify these effects (Coovadia et al., 2009). Disability grants for chronic schizophrenia often support entire households and may not offset healthcare costs (Burns and Esterhuizen, 2008). Apartheid-era planning concentrated on specialist services in urban centres, while many service users reside in under-resourced townships and peri-urban areas distant from tertiary care. High transport costs, unsafe travel and fragmented primary care integration impede sustained engagement. Income inequality has been linked to poor treatment of first-episode psychosis (Burns and Esterhuizen, 2008), emphasising structural contributions to relapse risk. Local evidence documents barriers to access and continuity (Bruwer et al., 2011). Our findings extend this literature, showing how spatial inequities heighten vulnerability through medication stockouts, premature discharge due to bed shortages and return to unstable, substance-exposed environments.

Healthcare fragmentation compounds these pressures. Early discharge, medication shortages and limited post-discharge rehabilitation disrupt continuity, especially for individuals whose symptoms impair help-seeking (Semahegn et al., 2020). Moncrieff et al. (2023) reported a 25% relapse rate over 24 months among individuals receiving reduced antipsychotic treatment versus 13% among those maintained on therapy, supporting the importance of continuity of care. In South Africa, most mental health funding is directed to inpatient care (Docrat et al., 2019), while primary care integration and community-based services remain limited (Baker and Naidu, 2020; Sorsdahl et al., 2020). Shortages of supervised community care and vocational rehabilitation further increase relapse risk. Clinicians

described discharge into crime, and substance-affected communities with rates of unemployment, where safety concerns, including gang violence in the Western Cape, often supersede health-seeking behaviour, resembling conflict settings yet underexplored in urban LMICs (Charlson et al., 2018). Cybersecurity breaches delaying laboratory results also impede care, which requires verified white cell counts before dispensing of Clozapine. Collectively, relapse appeared less as a discrete clinical event than the cumulative effect of systemic failures, although each was individually manageable, yet overwhelming in combination for clinicians and service users alike.

At the family level, financial strain, competing demands and limited psychoeducation constrained sustained support (McFarlane et al., 2003). Economic desperation sometimes led to illicit activities, creating conditions in which relapse is almost inevitable. Community and healthcare stigma further undermined engagement. Misconceptions framing schizophrenia as dangerous or a personal weakness contribute to discrimination (Hugo et al., 2003; Corrigan, 2004). Service users reportedly felt devalued when their concerns were minimised. In overburdened systems, chronic mental illness is often deprioritised relative to acute conditions, eroding trust (Blanchard and Lurie, 2004). Community stigma delays help-seeking, and when individuals do access services, experiences of deprioritisation within healthcare settings risk reinforcing avoidance. Clinicians consistently attributed relapse to structural, social and individual factors, with limited self-implication, which likely reflects high caseloads, resource constraints and limited reflective practice. These accounts represent perspectives shaped by professional experiences and institutional context, not causal explanations.

Although psychosocial interventions reduce relapse risk (Bighelli et al., 2021), addressing environmental stressors, housing instability, transport barriers and community safety remains essential in LMICs. What is striking across these accounts, clinicians frequently framed relapse as structurally inevitable rather than clinically preventable, reflecting both working conditions and biomedical training limits in addressing social determinants. From a policy perspective, relapse prevention requires strengthening primary mental healthcare, ensuring reliable medication supply, expanding supervised care and vocational programmes, implementing stigma reduction and improving community safety. This necessitates intersectoral collaboration across health, housing, social development and safety sectors, alongside collaborative care models (Myers et al., 2019; Fitts et al., 2020). Policies enabling cross-sector linkages, sustainable community service financing and redressing spatial inequities are critical.

Limitations include a focus on public-sector clinicians in LMIC communities, limiting transferability. The predominantly female, hospital-based sample showed no observed gender differences, though this warrants caution. Manual coding supported close engagement, consistent with reflexive thematic analysis, but lacked automated audit trails and visual mapping. Nonetheless, the qualitative design yielded rich insights into systemic barriers, with reflexive thematic analysis supporting depth and rigour. As the first phase of a multi-stakeholder study, these findings inform future research incorporating service users, families and administrators to develop contextually relevant, system-level relapse prevention strategies.

## Conclusion

Clinicians in South Africa identified medication non-adherence as the most common trigger for relapse in schizophrenia but framed it within interconnected structural challenges, including poverty, substance misuse, transport barriers, medication shortages, fragmented

care, stigma and community violence. Effective relapse prevention requires integrated strategies combining clinical management with interventions addressing these structural determinants. Policy shifts to strengthen primary care, ensure medication availability, expand rehabilitation and foster intersectoral collaboration are essential for sustained impact. Future research will incorporate perspectives of clients, families and administrators to inform feasible, contextually grounded relapse prevention approaches.

**Open peer review.** To view the open peer review materials for this article, please visit http://doi.org/10.1017/gmh.2026.10200.

**Supplementary material.** The supplementary material for this article can be found at http://doi.org/10.1017/gmh.2026.10200.

**Data availability statement.** The authors confirm that the data supporting the findings of this study are available within the article and its Supplementary Materials.

**Acknowledgements.** The authors would like to thank the clinicians for participating in this research. The authors would like to acknowledge Warona Mateane (WM) for her assistance with the reflexive thematic analysis.

**Author contribution.** Smit Retha: Writing – original draft, project administration, methodology, investigation, transcription and thematic analysis, coding of data, data conceptualisation. Luckhoff Hilmar: Writing – Review and editing. Phahladira Lebogang: Writing – review and editing. Kilian Sanja: Writing – Data conceptualisation, review and editing. Emsley Robin: Writing – review and editing. Asmal Laila: Writing – review and editing, validation, supervision, methodology, investigation, data conceptualisation.

**Financial support.** Funding to collect data for this study was provided through the University of Stellenbosch, Faculty of Medicine and Health Sciences, Department of Psychiatry.

**Competing interests.** The authors declare none.

**Ethics statement.** Ethical approval was obtained from the Health Research Ethics Committee of Stellenbosch University (Ethics reference number: N23/11/143). Participants were informed of the study objectives, assured that participation was voluntary and provided written informed consent. An anonymised identification number ensured confidentiality during transcription, analysis and publication.

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
