## [Reviewer Report]

Comments to the Author

This manuscript makes a contribution to understanding schizophrenia relapse in lower-middle-income country (LMIC) contexts. Several issues are worthy of in-depth exploration.

1.Interview Guide: The manuscript mentions a “semi-structured interview guide” informed by social constructivist theory but does not provide examples of core questions or probes. Including a brief appendix or summarizing key questions would help readers evaluate how the guide elicited insights on structural and social determinants.

2.Reflexivity: The first author’s reflexive statement is thoughtful, but expanding on how reflexivity shaped data analysis (e.g., how biases from working in a well-resourced academic setting were mitigated during coding or theme development) would further enhance transparency. For instance, did the research team include members with public-sector experience to balance perspectives?

3.Qualitative Analysis Tool Limitation: The current analysis relies solely on Microsoft Word for transcription and manual coding, without using specialized qualitative analysis software (e.g., NVivo). Such software enables systematic coding management, thematic visualization, inter-coder reliability testing, and saturation verification—key features that enhance the rigor and transparency of qualitative research. If specialized software was not used, the limitations of manual analysis should be acknowledged, and future improvements suggested.

4.Insufficiency of Current Tables: The manuscript only includes two tables (theme classification and participant demographics), which are insufficient to visually represent the complex interactions between variables. Adding figures to visualize core findings would help readers quickly grasp the study’s logic and key conclusions.

5.Subtheme Elaboration: Some subthemes would benefit from deeper analysis or disaggregation:

•Under “Healthcare system shortfalls,” “fragmented services” could be unpacked to specify gaps (e.g., poor care coordination between inpatient and outpatient settings, lack of discharge planning protocols).

•Under “Stigma,” distinguishing between community stigma (e.g., name-calling) and healthcare setting stigma (e.g., deprioritization) is valuable, but further exploration of their interaction (e.g., how healthcare stigma reinforces care avoidance) would add depth.

6.Demographic Difference Analysis: The sample is predominantly female (78.6%) and hospital-based (78.6%). Were there gender differences in clinicians’ perspectives? Did community clinic-based clinicians report distinct challenges compared to hospital-based counterparts? A brief analysis of these differences would address potential sample biases .

7.The discussion briefly links findings to broader LMIC literature but could more explicitly embed results within South Africa’s unique landscape. For example: How do spatial inequities specifically shape access to mental health services and relapse risk? How do the findings align with or diverge from previous South African studies ?

I strongly recommend that the authors enhance the manuscript by employing specialized software for qualitative interview data analysis, refining the coding process, and developing exemplary figures to clearly illustrate the logical relationships among variables. To elaborate, specialized qualitative analysis software (e.g., NVivo) should be adopted to replace the current manual coding approach relying on Microsoft Word. This software enables systematic management of coding nodes, visualization of thematic connections, and rigorous verification of inter-coder reliability and thematic saturation. Such improvements will significantly strengthen the methodological rigor, transparency, and reproducibility of the qualitative analysis. Furthermore, supplementing the manuscript with exemplary figures is critical to better communicate complex relationships. These figures will transform abstract textual descriptions into intuitive visual summaries, enhancing the readability and accessibility of core findings for readers, including policymakers and clinical practitioners.

---

## [Reviewer Report]

CLINICIANS’ PERSPECTIVES ON STRUCTURAL AND SOCIAL DETERMINANTS OF SCHIZOPHRENIA RELAPSE IN SOUTH AFRICA’S PUBLIC MENTAL HEALTH SERVICES

Journal: Cambridge Prisms: Global Mental Health

Manuscript ID: GMH-2025-0438

Reviewer comments

The authors present an exploratory qualitative study on clinicians’ perspectives regarding factors associated with relapse in schizophrenia within the South African context. Their objective focused on structural and social contributors to relapse. They identified three main themes and seven subthemes. The proposed framework seeks to conceptualise schizophrenia relapse beyond pharmacological non-adherence, including other factors such as health-service deficits, social determinants of mental health, violence, crime, stigma, and family characteristics, among others.

Qualitative studies are increasingly present in the medical literature; however, readers may not always be familiar with key methodological aspects of this type of research. In addition, researchers can make efforts to present results in more accessible and engaging ways to support understanding of the findings and the authors’ reflections. Simple tools, such as a summary diagram presenting the main themes and subthemes and their relationship to relapse and/or adherence problems, may facilitate readers’ retention of the paper’s key messages.

Considering the reflexivity described in Section 2.5, several questions came to mind while reading the draft: What, specifically, is new or different in the information elicited from the focus group(s)? Are the experiences of this group of clinicians similar to, or different from, those of clinicians in other LMICs? What reflections emerged regarding the gap between available evidence—much of which comes from the Global North—and the implementation of mental health services for people with a diagnosis of schizophrenia?

I’ll contribute with comments that could help to the authors to improve their proposal.

1. Tittle and general aim

The proposed title does not include any reference to the methodology used. Because qualitative studies remain less common in medical journals, please consider including some reference to the methodology in the title.

The authors briefly present the association between relapse and non-adherence to antipsychotic medication. I suggest considering a shift in focus from relapse to non-adherence to treatment—in the title, objectives, and throughout the draft—to reinforce the invitation to readers to adopt a broader understanding of this phenomenon, consistent with the themes and subthemes presented. In fact, this aligns with the authors’ own statement: “This study explores clinicians’ perspectives on relapse in schizophrenia in South Africa. It aimed to move beyond the dominant focus on non-adherence to examine broader contributing factors in resource-constrained settings”.

2. Impact Statements

In the statement “By revealing how social and structural factors interact with clinical care, this study contributes to global efforts to build more equitable, integrated, and contextually appropriate mental health services”, the authors may be overestimating what can be concluded from this study. The methodology used helped to elicit important information but does not explore or propose hypotheses about mechanisms underlying the interaction between clinical and social/structural factors. Please consider revising this statement accordingly.

3. Introduction

Consider including more information specifically on non-adherence to treatment in schizophrenia (and not only factors related to relapse). Because clozapine was mentioned by focus group participants, consider also including some specific information regarding that antipsychotic.

A very recent systematic review and meta-analysis of non-adherence to treatment in schizophrenia in Sub-Saharan Africa may help the authors enrich this section:

Addisu ZD, Demsie DG, Tafere C, Siraj EA, Yazie TS, Yimer EG, Alemu NG, Milikit YZ, Wabela BN, Yismaw MB, Ayal MA, Beyene DA. Non-adherence with the treatment regimen and its associated factors among patients with schizophrenia in Sub-Saharan Africa: a systematic review and meta-analysis. Sci Rep. 2025 Oct 29;15(1):37843. doi: 10.1038/s41598-025-21647-6. PMID: 41162442; PMCID: PMC12572252.

4. Methods

Did one or more authors participate in one of the three focus groups? If so, this should be mentioned in Section 2.1.

In Section 2.4, the text states: “Recordings were transcribed using Microsoft Word and verified for accuracy by RS and WM”. The initials WM do not correspond to anyone in the author list. Please review and correct this throughout the manuscript.

The definition of themes and subthemes is a key issue in this type of qualitative study. In Section 2.4, consider including a more detailed description of how the authors reached consensus on the definitions of these items.

Section 2.5: Reflexivity is presented for the lead author (RS). Were similar reflexivity exercises conducted for the remaining authors? If so, it would be important to include some information about this.

5. Results.

Demographic data of participants: it seems important to include participant demographics in the main text. Consider reporting metrics related to years of clinical experience across participant groups. Information on individual caseload (where available) could also support a clearer understanding of the interviewed professionals.

Main themes and subthemes: as mentioned above, consider the possibility of shifting from relapse to non-adherence. With this change, a theme could be, for example, “Individual and treatment-related contributors to non-adherence”. Given the nature of qualitative research, I strongly suggest discussing this proposal with the full author team before making any change. Please do not accept this reviewer’s suggestion immediately.

The subtheme “Substance use and peer pressure” includes citations mainly focused on substance use. The peer pressure component is not clear; consider adding one or two citations to clarify this aspect of the subtheme.

In the “Crime and unsafe communities” subtheme, do the authors have additional citations to complement the description of clinicians’ perceived effects of environmental violence on people with schizophrenia in South Africa? This could help readers better understand how this may impact adherence to antipsychotics.

6. Discussion

The authors provide a helpful discussion of each theme and subtheme, with appropriate details and ad hoc references.

If the authors accept the suggestion made above about using the concept of non-adherence to treatment instead of relapse, corresponding changes should be made in the Discussion to maintain consistency.

I invite the authors to consider recent views and discussions about discontinuation of medication in people with a diagnosis of schizophrenia. Some important references are:

Moncrieff J, Crellin N, Stansfeld J, Cooper R, Marston L, Freemantle N, Lewis G, Hunter R, Johnson S, Barnes T, Morant N, Pinfold V, Smith R, Kent L, Darton K, Long M, Horowitz M, Horne R, Vickerstaff V, Jha M, Priebe S. Antipsychotic dose reduction and discontinuation versus maintenance treatment in people with schizophrenia and other recurrent psychotic disorders in England (the RADAR trial): an open, parallel-group, randomised controlled trial. Lancet Psychiatry. 2023 Nov;10(11):848-859. doi: 10.1016/S2215-0366(23)00258-4. Epub 2023 Sep 28. Erratum in: Lancet Psychiatry. 2023 Nov;10(11):e29. doi: 10.1016/S2215-0366(23)00342-5. PMID: 37778356.

Special feature on antipsychotic discontinuation in schizophrenia from the journal Schizophrenia, available at: https://www.nature.com/collections/dheeaebgfj

---

## [Reviewer Report]

CLINICIANS’ PERSPECTIVES ON STRUCTURAL AND SOCIAL DETERMINANTS OF SCHIZOPHRENIA RELAPSE IN SOUTH AFRICA’S PUBLIC MENTAL HEALTH SERVICES

General comments

I found this article very interesting as it addresses an important issue, which is relapse in people living with schizophrenia, which has a major impact at personal, family and social levels.

The authors present this study as the first phase of a multi-stakeholder study to address the issue. They adopt a qualitative approach to explore clinicians’ perspectives related to structural and social determinants. This is highly innovative and relevant because the issue has been mostly studied from a quantitative perspective and has focused mainly in clinical factors.

Specifically, the authors use Reflexive Thematic Analysis (RTA) as a framework for analyzing data collected in three focus groups. The results showed that from the clinicians’ perspective, medication non-adherence is the most important cause of relapse and that this is framed within intersecting challenges rather than individual failings. This was described in three overarching themes: (i) individual and treatment-related contributors; (ii) structural- and environmental-level contributors; and (iii) social- and community-level contributors.

I think this study has the potential to contribute to clinical practice and research and, in my view, it should be accepted with major revisions. This is mainly because some issues regarding the alignment with RTA principles and qualitative reporting would benefit from further clarification, as outlined below. RTA, situated within a “Big Q” approach to qualitative research, considers subjectivity as a key source of knowledge and, what might be considered a bias in quantitative methods is precisely what enriches this perspective (Braun & Clarke, 2025). While I do not work primarily with RTA, my understanding is that it carries specific conceptual and reporting requirements that demand closer attention.

Introduction

It would be interesting to consider a deeper account of the personal and institutional contexts in which this study was conceived and conducted, including a description of how mental healthcare for people living with schizophrenia is organized in South Africa. I would also suggest citing the recently published consensus on relapse in schizophrenia (Howes et al., 2025) which emphasizes that clinical judgement is still the most utilized criteria, underlining the subjective dimension of clinical decision-making, and therefore, the need for a qualitative approach to the topic.

Methodology

I would suggest that the section on “Reflexivity” could be renamed “Research positioning” and, if it makes sense to the authors, to be placed before data analysis, as for example, in Graham & Clarke (2021). I think this would better show how reflexivity was part of the whole research process and it would help the reader to understand how this positioning informed the analytic process. The existing description is very interesting and is headed in the right direction. However, it would greatly benefit from a clearer account of which specific events and discussions moved the author to engage with this topic.

A second issue has to do with the “reconciled discrepancies” during coding. In RTA, differences in interpretation are often treated as an analytically rich resource rather than as errors or bias. I would suggest revising the description of team involvement to better reflect an RTA-consistent approach and describing how divergent readings were handled. Showing how differences contributed to theme development may even strengthen the manuscript.

Results

The section is clearly structured, and the three overarching themes are easy to follow. That said, I think the manuscript would benefit from a more developed building of each theme. In RTA papers, authors tend to take more space to unfold the central meaning that organises each theme and to show how the interpretation was generated.

In the manuscript, themes sometimes appear rather quickly, with limited room for the analysis that linked extracts to the interpretation. If feasible within word limits, I would suggest expanding the analytical development of each theme, and to ideally add more extracts that support each argument.

Discussion

In general, this section reads more like a deepening of the introduction than a discussion of results. It also contains a few new ideas that do not clearly connect with the presented results, as for instance: “For some individuals with schizophrenia, psychosis itself may become a source of companionship or empowerment, in the absence of alternative sources of meaning and social connection”. I would suggest linking this more clearly with the data.

I would also suggest strengthening this section by a more explicit analysis of the perspective of clinicians’ accounts. At times, clinicians’ perspectives appear to be taken at face value as “facts” that directly explain relapse. I think that these accounts might be analyzed as viewpoints that are structured, for instance, by professional identity or institutional constraints. Furthermore, while clinicians frequently emphasize social and structural failures, it might be also considered the relative absence of self-implication or reflection on clinicians’ own role in this phenomenon. Perhaps, the authors might also consider adding a brief gender perspective reflection, considering the composition of sample

Finally, a further issue concerns the use of generalizability language, as generalization is typically not the aim in qualitative research. Rather, notions such as transferability are more appropriate and depend on rich contextualization. This reinforces the need for more detailed description of the clinical, organizational, and relational contexts in which clinicians’ perspectives were produced and interpreted.

I hope these comments will be helpful and I thank for the opportunity of making this review. I apologize to the authors and editor for any misunderstandings or factual errors I may have made in reading this manuscript.

References

Braun, V., & Clarke, V. (2025). Reporting guidelines for qualitative research: A values-based approach. Qualitative Research in Psychology, 22(2), 399–438. https://doi.org/10.1080/14780887.2024.2382244

Graham, R., & Clarke, V. (2021). Staying strong: Exploring experiences of managing emotional distress for African Caribbean women living in the UK. Feminism & Psychology, 31(1), 140–159. https://doi.org/10.1177/0959353520964672

Howes, O. D., Bukala, B. R., Chen, E. Y. H., Correll, C. U., Hasan, A., Honer, W. G., Kane, J. M., Leucht, S., Siafis, S., Agid, O., Akena, D., Arango, C., Atwoli, L., Barnes, T. R. E., Birnbaum, M. L., Bitter, I., Breier, A., Buchanan, R. W., Citrome, L., … McCutcheon, R. A. (2025). Relapse in Schizophrenia: A Systematic Review of Criteria for Clinical Studies and International Consensus Guidelines to Improve Them. American Journal of Psychiatry, 182(11), 969–983. https://doi.org/10.1176/appi.ajp.20241040

---

## [Editor Report]

Dear Mrs. Smit,

Folllowing careful review from three independent reviewers, we can inform you that the decision on your submission is “Major Revision”. Please respond systematically to each query, and kindly indicate how each point of critique has been resolved. We look forward to receiving your revised manuscript. 

Best regards,

Prof. André Janse van Rensburg

Cambridge Prisms: Global Mental Health

Manuscript ID: GMH-2025-0438

Manuscript Type: Research Article

Manuscript Title: CLINICIANS’ PERSPECTIVES ON STRUCTURAL AND SOCIAL DETERMINANTS OF SCHIZOPHRENIA RELAPSE IN SOUTH AFRICA’S PUBLIC MENTAL HEALTH SERVICES

Site URL: https://mc.manuscriptcentral.com/prisms-gmh

---

## [Reviewer Report]

Authors have addressed in detail the different comments of the reviewers. They incorporated most of the suggestions made and justified in an appropriate way the suggestions that were rejected. I think the new version of the manuscript is ready to be published. My only suggestion is to apply some colours to Figure 1 to improve its design. Congratulations to the authors.

---

## [Reviewer Report]

I would like to thank the authors for addressing the issues observed during the first revision. The article has greatly improved in all its sections. I would only suggest a minor optional modification in the figure title to make it clear that this was built over clinician’s perspectives.

Having said that, I am pleased to recommend that this article is accepted for publication.

---

## [Editor Report]

Dear Mrs. Smit,

Following your revisions in response to peer review critiques and inputs, we have the pleasure to inform you that your paper has been accepted for publication. The editorial office will be in touch with next steps shortly.

Sincerely,

Prof. André Janse van Rensburg

Cambridge Prisms: Global Mental Health

Manuscript ID: GMH-2025-0438.R1

Manuscript Type: Research Article

Manuscript Title: BEYOND TREATMENT NON-ADHERENCE: A QUALITATIVE STUDY OF CLINICIANS’ PERSPECTIVES ON STRUCTURAL AND SOCIAL DETERMINANTS OF SCHIZOPHRENIA RELAPSE IN SOUTH AFRICA

Site URL: https://mc.manuscriptcentral.com/prisms-gmh